# Retrospective Study: Lateral Ridge Augmentation Using Autogenous Dentin: Tooth-Shell Technique vs. Bone-Shell Technique

**DOI:** 10.3390/ijerph18063174

**Published:** 2021-03-19

**Authors:** Michael Korsch, Marco Peichl

**Affiliations:** 1Dental Academy for Continuing Professional Development, Karlsruhe, Lorenzstrasse 7, 76135 Karlsruhe, Germany; marco_peichl@za-karlsruhe.de; 2Clinic of Operative Dentistry, Periodontology and Preventive Dentistry, University Hospital, Saarland University, Building 73, 66421 Homburg, Germany; 3Private Practice, Center for Implantology and Oral Surgery, Berliner Str. 41, 69120 Heidelberg, Germany

**Keywords:** tooth-shell technique, implant, dentin, bone graft, autogenous

## Abstract

In the literature, autogenous dentin is considered a possible alternative to bone substitute materials and autologous bone for certain indications. The aim of this proof-of-concept study was to use autogenous dentin for lateral ridge augmentation. In the present retrospective study, autogenous dentin slices were obtained from teeth and used for the reconstruction of lateral ridge defects (tooth-shell technique (TST): 28 patients (15 females, 13 males) with 34 regions and 38 implants). The bone-shell technique (BST) according to Khoury (31 patients (16 females, 15 males) with 32 regions and 41 implants) on autogenous bone served as the control. Implants were placed simultaneously in both cases. Follow-up was made 3 months after implantation. Target parameters during this period were clinical complications, horizontal hard tissue loss, osseointegration, and integrity of the buccal lamella. The prosthetic restoration with a fixed denture was carried out after 5 months. The total observation period was 5 months. A total of seven complications occurred. Of these, three implants were affected by wound dehiscences (TST: 1, BST: 2) and four by inflammations (TST: 0, BST: 4). There were no significant differences between the two groups in terms of the total number of complications. One implant with TST exhibited a horizontal hard tissue loss of 1 mm and one with BST of 0.5 mm. Other implants were not affected by hard tissue loss. There were no significant differences between the two groups. Integrity of the buccal lamella was preserved in all implants. All implants were completely osseointegrated in TST and BST. All implants could be prosthetically restored with a fixed denture 5 months after augmentation. TST showed results comparable to those of the BST. Dentin can therefore serve as an alternative material to avoid bone harvesting procedures and thus reduce postoperative discomfort of patients.

## 1. Introduction

Autogenous bone is still considered to be the gold standard for the reconstruction of lateral alveolar ridge deficits due to its excellent osteoconductive, osteoinductive, and osteogenetic properties [1]. The disadvantage, however, is that in many cases a donor region is necessary.

Recent research has focused on dentin as an alternative autogenous grafting material because of its similarity to the bone in organic and inorganic compositions [2,3,4]. The proportion of inorganic substances in human dentin is approximately 69% and the proportion of organic components is approximately 17.5%. The alveolar bone consists of approximately 62% inorganic components and 25% organic components. In particular, the organic matrix of dentin and bone mainly consists of type I collagen (~90%) and it contains many noncollagenous structural proteins such as osteocalcin, osteonectin, phosphoprotein, and sialoprotein, as well as osteogenetic growth factors—i.e., bone morphogenic proteins (BMPs), tissue growth factor-ß (TGF-ß) or insulin like growth factor-2 (IGF-2) [5,6].

As in alveolar bone, the inorganic components of dentin are primarily composed of various calcium phosphates (hydroxyapatite, ß-tricalcium phosphate, octacalcium phosphate, and amorphous calcium phosphate) [7]. Due to their good osteoconductive properties, these components are used as alloplastic bone substitute materials. The osteoconductive and inductive properties of dentin, which can promote bone formation at grafted defect sites, have been demonstrated in several animals and human clinical studies [4,8,9,10,11,12,13,14]. There is significantly less resorption of the graft compared to autogenous bone grafts—e.g., monocortical bone block grafts from the retromolar region [15,16,17,18]. Other outcome measures based on histological, immunohistochemical, and radiologic evaluations did not show significant differences between dentin and bone grafts [19,20]. There is histological evidence that grafts derived from autogenous tooth material are involved in bone remodeling processes and allow sufficient osseointegration of dental implants [15,21,22]. Even though teeth that were in contact with the oral cavity were used for grafting, no increased inflammation could be observed histologically [16] and there was no increased level of wound infections or loss of grafts [23].

The tooth-shell technique (TST) used in this study is intended for lateral ridge augmentation [24]. This technique is a modification of the bone-shell technique (BST) described by Khoury [25]. The bone-shell technique using cortical bone obtained from the linea obliqua combines the stability of cortical bone grafts with improved osteoconductive properties. With this technique, a stable scaffold will be created by the thin cortical bone shell, which is rigidly fixed at a distance. The resulting gap is filled with autogenous bone particulate. Both the cortical bone shell and the autogenous bone particulates promote the revascularization and the regeneration of the graft [25]. As a result of the structural and chemical similarities of dentin and alveolar bone, equally good results can be expected for the procedure using a dentin shell and particulate dentin. The main difference between the two techniques is that with dentin grafts, bone harvesting and the possible resulting donor site morbidity can be avoided. The technique used to prepare the dentin graft is a commercially available technology consisting of a grinder for particulation of the dentin and substances for disinfection and demineralization. The advantage of this technology is that the dentin can be prepared chair-side and is immediately available for grafting. Other commercially available autogenous tooth-derived grafting materials, e.g., the autogenous demineralized dentin matrix (ADDM, AutoBT), are based on processing of the dentin under industrial conditions in a tooth bank. However, this procedure is only available in certain geographical regions.

The present retrospective study is a proof-of-concept study. This study aimed to determine whether the use of autogenous dentin is suitable for the reconstruction of lateral ridge defects. If the technique is suitable for lateral ridge augmentation, postoperative discomfort could be reduced in the future in cases with teeth that cannot be preserved compared to bone block augmentation.

## 2. Materials and Methods

For this proof-of-concept study, cases for lateral ridge augmentation were re-examined in which autogenous dentin (TST) with simultaneous implantation was used between 1 June 2019 and 31 March 2020. Patients in whom BST with autogenous bone was used during the same period served as the control group.

In cases in which a tooth suitable for grafting was present (e.g., a non retainable tooth or an impacted wisdom tooth), the TST was used. In cases with absence of a suitable tooth, the BST was carried out.

The electronic medical records of the individual patients were used to screen for potential cases for inclusion in this study. All surgical procedures were carried out by an experienced oral surgeon (MK). The study protocol was approved by the Institutional Review Board of the Baden-Württemberg Medical Board (ID: F-2020-068-z). The study was conducted according to the principles of the Declaration of Helsinki and the EQUATOR guidelines. The inclusion criteria were defined as described below:**Inclusion criteria:**
-age > 18 years;-alveolar crest augmentation of a lateral bony defect with autogenous dentin (TST) or autogenous bone (BST);-lateral alveolar crest defect of at least 4 mm in the region of implant placement prior to augmentation;-restauration with a fixed denture is provided;-edentulous region of maximum two missing teeth.
**Exclusion criteria:**
-age < 18 years;-untreated or residual periodontitis;-uncontrolled diabetes mellitus with HbA1c >7%;-malignant neoplasm;-history of bisphosphonates and/or radiotherapy in region of head and neck;-immunosuppression or immunosuppressant therapy;-lateral alveolar crest defect of less than 4 mm in the region of implant placement prior to augmentation;-restauration of the implant with a removable denture is intended.

In all of the cases (TST and BST), the width of the bucco-palatal bone was measured with a preoperative cone-beam computed tomography (CBCT) before augmentation. At least 1.5 mm of bone/autogenous dentin should cover the implants on the buccal and palatal surfaces. The achieved ridge width was a result of the desired implant diameter—a 1.5 mm buccal and 1.5 mm oral lamella of original bone or autogenous bone/dentine graft. A ridge width of at least 6.8 mm was aimed for when the implant diameter was 3.8 mm. A gain of hard tissue of at least 4 mm was an augmentation procedure requirement in all cases.

The patients were divided into the following two groups:Group 1 (control group)Bone-shell technique (BST): 31 patients (16 female, 15 male) with 32 regions and 41 implants.

Lateral ridge augmentation was carried out using an autogenous bone graft from the retromolar area of the mandible. A thin bone slice, which was obtained from the bone block, was fixed in the region of augmentation with osteosynthesis screws according to Khoury [25]. The cavity between the bone slice and the original bone was filled with particulate autogenous bone.


*Group 2*
Tooth-shell technique (TST): 28 patients (15 female, 13 male) with 34 regions and 38 implants.

Lateral ridge augmentation was carried out using an autogenous dentin slice with osteosynthesis screws and particulate dentin according to the BST.

The following data were extracted from the electronic medical record:-demographic data: age and gender;-data on restoration and data on maintenance therapy;-complications: loss of graft, loss of implant, dehiscences, infections/inflammations, nerve injuries;-implant data: type, length/diameter and region.

The analyzed target parameters were:-clinical complications;-peri-implant bone loss;-osseointegration;-integrity of the buccal lamella.

### 2.1. Clinical Complications

All complications concerning the graft or the implant that occurred during the follow-up period were noted. The loss of the graft either through infection or unexpected massive resorption and the loss of an implant during the follow-up period were defined as severe complications. Dehiscences, transient nerve injuries and infections/inflammation of the grafted site were categorized as non-severe complications if the implant was fully osseointegrated.

### 2.2. General Surgical Procedures

Perioperative antibiosis (one day pre- and two days postoperatively) with amoxicillin (750 mg) three times per day was prescribed. For patients with penicillin intolerance, clindamycin (300 mg) was used. As analgesic Ibuprofen (400 mg) was prescribed and used as required. For all surgical procedures local anesthesia was performed using articaine with epinephrine 1:100.000 (Citocartin Sopira^®^, Heraeus Kulzer GmbH, Hanau, Germany).

For exposure of the defect site a full-thickness flap with a crestal incision and mesial or distal releasing incision was raised in both treatment groups. The implant bed was prepared according to the protocol of the implant manufacturer and the implants were inserted at the hard tissue level. To ensure a sufficient soft-tissue closure without tension, the flap was mobilized with a periosteal releasing incision. Non resorbable sutures (Supramid^®^ 5/0, Serag Wiessner, Naila, Germany) were used for wound closure. A two-dimensional radiograph (panoramic X-ray) was taken to assess the surgical procedure.

#### 2.2.1. Surgical Procedure of the BST

In the the BST group, a cortical bone block was harvested from the region of the posterior mandible (retromolar region/linea obliqua) with the MicroSaw-Kit^®^ (Dentsply Friadent, Mannheim, Germany) (Figure 1a). Two thin bone slices were obtained by longitudinal splitting of the bone block with a diamond disc (Komet^®^ Gebr. Brasseler, Lemgo, Germany) (Figure 1b,c). One slice was fixed laterally of the alveolar ridge defect with osteosynthesis screws (MicroScrew^®^-Kit, stoma^®^Instruments, Tuttlingen, Germany) (Figure 1d,e). The residual bone was particulated with a bone crusher (stoma^®^Instruments, Tuttlingen, Germany). In addition, autogenous bone chips were obtained with a twist drill (Pilot Drill 1, length 41 mm, Ø 2.2 mm, Straumann Group, Basel, Switzerland) by a screen hole drilling technique in the posterior mandibular region. The cavity between the fixed bone slice and the alveolar bone was filled with the mixture of the autogenous bone chips (Figure 1f). Membranes and/or bone substitute materials were not used for this technique. Wound closure was achieved as described above.

#### 2.2.2. Clinical Procedure of the TST

Immediately after extraction, the tooth that was intended to be used for augmentation was cleaned mechanically by removing debris and the periodontal ligament as well as restorations and root filling material with a coarse diamond bur under water cooling (Figure 2a). A slice of root dentin of 1–1.5 mm thickness was obtained with a diamond disc (Frios MicroSaw, Dentsply Sirona Implants, Mannheim, Germany) under water cooling (Figure 2b). The remaining tooth structure was particulated with a sterile disposable grinder (Smart Dentin Grinder, Kometa Bio, Creskill, USA) to 300–1200 μm particles (Figure 2c,d). The dentin shell and the particulate dentin underwent a chemical cleaning procedure by being placed in a sterile closed dappen dish with a solution of sodium hydroxide (0.5 N, 4 mL) and ethanol (20 Vol.%, 1 mL) (Dentin Cleanser, Kometa Bio, Creskill, NJ, USA) for 10 min. Subsequently, the supernatant was absorbed with sterile gauze and the material was cleaned additionally for another 3 min by manual shaking in phosphate-buffered physiological saline solution (Dulbecco’s Phosphat Buffered Saline, Kometa Bio, Creskill, NJ, USA). A partial demineralization of the dentin for exposure of the collagen fiber network and release of osteoinductively active growth factors [10] was achieved by placing the material in a 10% EDTA solution for 3 min (EDTA solution, Kometa Bio, Creskill, NJ, USA). The material obtained was then cleaned once more with a buffered saline solution. Then, it was used immediately for grafting and/or dried at a moderate temperature (below 38 °C) on a hotplate and stored in a sterile vessel at −18 °C until grafting.

#### 2.2.3. Surgical Procedure of the TST

For this technique, the dentin slice was fixed laterally to the alveolar crest defect with osteosynthesis screws (Figure 3a) according to the BST. The cavity between the fixed dentin slice and the alveolar bone was filled with the crushed autogenous dentin particles (Figure 3b). Bone substitute materials were not used. The wound was closed as described above. For the TST, no bone substitute materials or membranes were used either.

#### 2.2.4. Implant Exposure

Three months after augmentation with simultaneous implantation, all implants (TST and BST) were exposed. Peri-implant hard tissue level assessment was performed by measuring from the implant shoulder to bone/hard tissue-implant contact with a periodontal probe at four locations (mesial, distal, oral and buccal). Implant stability measurement was carried out by a resonance frequency analysis in all cases (Ostell Idx, W&H, Buermoos, Austria). Implants with an implant stability quotient (ISQ) above 60 were approved for prosthetic restoration.

#### 2.2.5. Radiographic Evaluation

At the time of implant exposure (3 months after augmentation), a CBCT was made to assess the osseointegration, the buccal lamella, and the horizontal hard tissue loss. All implants were placed at the hard tissue level and the implant surfaces were completely covered by native bone or hard tissue graft (autogenous dentin or bone). To evaluate the horizontal bone loss the CBCTs at the time of implant exposure were analyzed. The hard tissue loss was assessed considering mesial and distal aspects. Only the highest value at the mesial or distal margin was included in the analyses. Additionally, the integrity of the buccal lamella was assessed using the CBCT (Figure 4a–d). Possible hard tissue loss with exposed buccal implant surfaces was noted.

#### 2.2.6. Osseointegration

Complete osseointegration was defined as:-no peri-implant bone/hard tissue loss > 1 mm at the four measuring points;-ISQ-Level > 60;-implant covered by a radio-opaque structure in CBCT;-integrity of the buccal lamella preserved in CBCT (no more than 1 mm loss).

#### 2.2.7. Prosthetic Restoration

The prosthetic restoration began four weeks after implant exposure. After a further 4 weeks, the dentures were incorporated, so that the treatment was completed after a total of 5 months.

### 2.3. Statistical Analyses

Data were compiled in Excel and analyzed with IBM SPSS Statistics 22 (SPSS Inc., Chicago, IL, USA) in Windows 7. The statistical methods included cross-tabulations with Fisher’s exact tests for categorical data. Mean values were compared by 2-sample t-tests.

All evaluations were analyzed at the patient, region and implant levels. For the evaluation at the region level, the different regions were distinguished as follows: grafts were assigned to different sextants independently of one another (more than two tooth widths apart from each other).

## 3. Results

Between 1 January 2019 and 31 March 2020, lateral ridge augmentation was performed in a total of 59 patients (31 females, 28 males) at 66 implant regions (Table 1). Simultaneously with the TST or BST, a total of 79 implants were inserted. The implant systems used were: ASTRA TECH Implant System ™ EV (Astra Tech Implant System, Dentsply Sirona, York, PA, USA), Nobel Biocare (Nobel Biocare, Kloten, Switzerland), and Conelog (CONELOG^®^, ALTATEC GmbH, Wimsheim, Germany). The mean age at the time of implantation was 61.2 years (Table 1). There were no significant differences concerning age and gender distribution between the two groups.

### 3.1. Total Number of Complications

The total number of clinical complications was lower in the TST group than in the BST group but there was no significance at a patient level, region level, and implant level (Table 2). At the implant level, seven clinical complications (TST: n = 1 vs. BST: n = 6) occurred. Of these seven complications, three implants were affected by wound dehiscences and four by inflammations. There were no significant differences between the two groups in terms of wound dehiscences at all three statistical levels. Only at the implant level was there significantly more inflammations for the BST. At the patient level and region level, there were no significant differences between the two groups in terms of inflammations. The wound dehiscences closed completely within after rinsing with 0.2% chlorhexidine-solution and the application of 1% chlorhexidine gel. Only in one case (BST) of dehiscence was a secondary suturing for sufficient wound closure performed after rinsing with chlorhexidine. Purulent infections were treated with local incision and drainage. Irrigations with chlorhexidine solution or sodium hypochlorite were then carried out daily until there was no longer any secretion. In these cases, an antibiotic was prescribed (amoxicillin (750 mg) or in case of penicillin intolerance clindamycin (300 mg), three times per day).

#### 3.1.1. Severe Clinical Complications

After the grafting procedures with simultaneous implant insertion and at the follow-up 3 months later, there were no severe clinical complications in both groups.

#### 3.1.2. Peri-Implant Tissue Probing

The probing depth did not exceed 0.5 mm for all implants.

#### 3.1.3. ISQ Values

The ISQ value was over 60 for all implants and was in the range of 61–89. The average ISQ value for BST was 74.7 and for TST this was 73.3. There were no significant differences between the groups.

### 3.2. Radiographic Evaluation

At the time of the follow-up, 3 months after augmentation with simultaneous implantation evaluation of the CBCTs showed two cases with horizontal hard tissue loss at the mesial or distal implant shoulder. One case with TST exhibited a hard tissue loss of 1 mm and one case with BST of 0.5 mm. There were no significant differences between the two groups at all three statistical levels. The integrity of the buccal lamella was preserved in all implants (Figure 4a,d). All implants were completely covered with hard tissue (no more than 1 mm loss).

In all cases with TST, the dentin shell was clearly visible. In some cases with BST, the bone shell was not visible in the CBCT or had a partial replacement resorption.

### 3.3. Osseointegration

Since there was no increased probing depth for any of the implants, the ISQ values were over 60, the integrity of the buccal lamella was preserved, and all implant surfaces were covered with hard tissue; all implants were, by definition, completely osseointegrated.

### 3.4. Prosthetic Restoration

All implants could be prosthetically restored with a fixed denture 5 months after augmentation. No complications occurred with any of the implants in the period between implant exposure and final prosthetic restoration.

## 4. Discussion

In this retrospective study, 28 patients with 34 regions and 38 implants in which the TST was applied were followed up. The BST according to Khoury served as a control group (31 patients, 32 regions, and 41 implants). The results showed that with regard to biological complications, horizontal hard tissue loss, osseointegration, and integrity of the buccal lamella, TST led to results comparable to those of BST. The use of autogenous dentin in TST, therefore, appears to be a possible alternative to autogenous bone.

For the reconstruction of alveolar crest defects before implant insertion, there are several available approaches—for example, guided bone regeneration techniques with the use of membranes and bone substitute materials such as xenografts or allografts. Many techniques are limited in terms of reconstructing three-dimensional defects. Autogenous bone block techniques are still considered to be the gold standard for this application because of the mechanical stability and the osteoconductive, osteoinductive, and osteogenetic properties of autogenous transplants [1]. The Khoury BST offers a procedure that allows even complex alveolar crest defects to be reconstructed with predictable results [25]. In comparison to other techniques of alveolar ridge reconstruction, BST has a low complication rate with a high prognosis for success [25,26,27]. Because of the reasons mentioned before and the similarity of the BST to TST, we considered the BST to be the ideal control group. Nevertheless, this technique is associated with possible complications, especially in the donor region, such as injuries of the inferior alveolar nerve or infections.

The tooth-shell technique described in this study is a variation of this technique using autogenous dentin instead of bone, which avoids bone harvesting procedures from retromolar mandibula or other donor sites. Recent research has shown that autogenous dentin shares many similarities with bone in its structural and chemical compositions and is, therefore, suitable as a bone substitution material with comparable biological properties and less resorption of the graft [17,18]. Autogenous dentin, however, has been shown to be involved in the remodeling process of bone and to be successively replaced by newly formed bone through replacement resorption more homogenously than bone grafts but leaving some remnants of the tooth material [16]. Partial demineralization of the dentin, which was performed by a 10% EDTA solution in this study, is able to promote the replacement resorption and new bone formation due to the exposure of the collagen network and the release of osteogenic growth factors—e.g., bone morphogenetic proteins. Completely demineralized dentin, on the other hand, is resorbed faster than new bone can be formed due to enzymatic degradation of the collagen network [28]. The clear visibility of the tooth shell in the CBCT image and the bone shell that is partially demarcable suggest a lower resorption rate for the TST than for the BST.

The lateral alveolar ridge augmentation techniques using autogenous dentin described in previous clinical studies primarily used the complete tooth root as a graft [17,23]. The dimension of the root, however, will limit the possible augmentation width. With the TST, a larger horizontal deficit can be augmented analogously to the BST, according to Khoury [25]. Furthermore, the particulate dentin in the gap between the tooth shell and bone defect can be expected to lead to better revascularization and bone regeneration than solid tooth roots as is described for bone block techniques with the use of particulate autogenous cortical bone [25].

In the present study, there were no severe complications (loss of graft and/or implant) with TST and BST. The implant survival rate was 100%. No nerve injuries with persistent or transient hyp- or paraesthesia caused by the harvesting procedure for BST occurred. The incidence of complications for both TST and BST are within the range of previous studies [1,17,18,23,25,26,29,30]. Wound dehiscence occurred in only three cases (BST: two (4.9%), TST: one (2.6%)). Compared to augmentation with Titan Mesh of up to 30%, the dehiscence rates in this study are very low [27]. Infections occurred in four cases of BST. These could all be treated without the loss of augmentation or implants.

A limitation of the present study is the lack of histological results. Both the BST and TST prevent a biopsy from being obtained in the case of simultaneous implantation. It is therefore not possible to assess whether the augmentation was integrated by new bone in TST and BST. The study, therefore, speaks of hard tissue. However, previous studies were able to demonstrate the integration of autogenous dentin in new bone formation [11,12]. Additionally, replacement resorption of dentin and contact between dentin augmentation and implant, which is comparable to an autogenous block graft, could be demonstrated histologically [16,31]. The low peri-implant probing depths during implant exposure and the radiological results of this study suggest this.

The radiographic at re-entry after 3 months showed a decent horizontal bone loss in only a few cases. The integrity of the buccal lamella was demonstrated in the CBCT for all implants. The peri-implant probing depths as well as the radiological results and the adequate ISQ values suggest osseointegration. Another limitation of the study is that a CBCT was only performed at the time of implant exposure. Therefore, the extent of resorption of the augmentation cannot be assessed in the observation period. In the case of the TST, however, a dentin shell was visible in the CBCT in all cases, while in BST the bone shell was not visible in some cases or apparently had a partial replacement resorption. Comparable studies show that the resorption of autogenous dentin is lower than that of bone [18].

A limitation of the study is the relatively short observation period of 3 months. However, the proof-of-concept study was able to demonstrate that, using the tooth-shell technique, implants can be completely osseointegrated if the bone supply is insufficient. Despite the limitations of the present study, the tooth-shell technique appears to be an alternative to bone block transplants. This could lead to avoiding donor regions for bone harvesting with increased postoperative discomfort in some cases. Further studies with comparative X-rays, histological examinations, and longer observation periods are recommended.

## 5. Conclusions

Within its limitations, this retrospective proof-of-concept study revealed that the tooth-shell technique represents a safe grafting procedure for lateral alveolar ridge augmentation with predictable results. Due to the avoidance of a second intervention for the harvesting of autogenous bone, the burden on the patient can be minimized.

## Figures and Tables

**Figure 1 ijerph-18-03174-f001:**
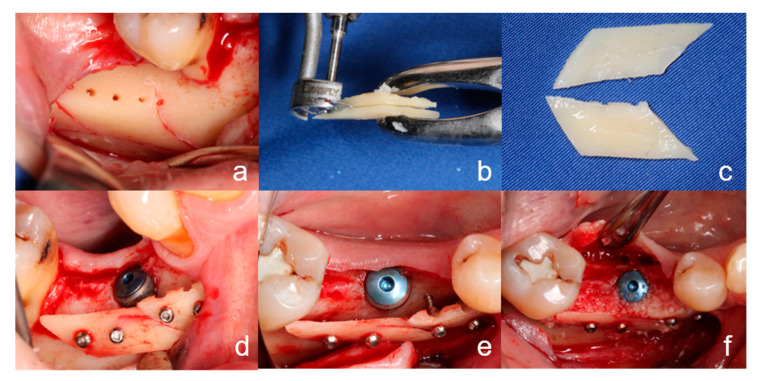
Clinical procedure of the bone-shell technique. (**a**): The graft was harvested with a MicroSaw-Kit^®^ from the retromolar region/linea obliqua. (**b**,**c**): The bone block graft was split into two thin bone slices with a diamond disc. (**d**,**e**): The thin bone slices were fixed at a distance from the alveolar ridge with osteosynthesis screws. (**f**): The cavity between the fixed bone slice and the alveolar ridge was filled with autologous bone chips.

**Figure 2 ijerph-18-03174-f002:**
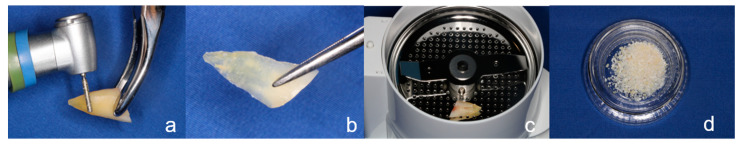
(**a**): The illustration shows the removal of debris and foreign material, such as restorations and root filling material, as well as the periodontal ligament, from the root surface with a coarse diamond bur under water cooling. (**b**): Dentin shell obtained from the root dentin with a diamond cutting disk. (**c**): Sterile disposable dentin grinder (Smart Dentin Grinder) for the particulates of dentin. (**d**): Particulate treated dentin.

**Figure 3 ijerph-18-03174-f003:**
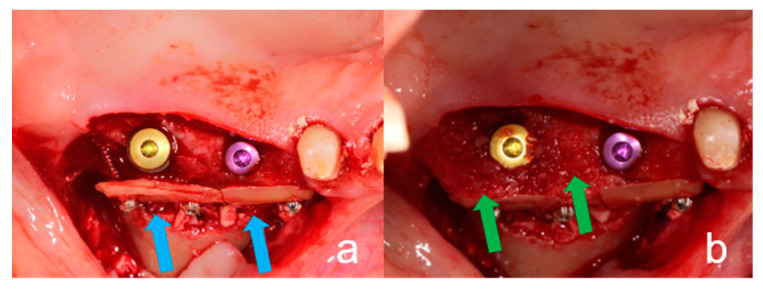
Clinical procedure of the tooth-shell technique. (**a**): Inserted implants at the site of tooth 25 and tooth 26 with vestibular bone missing. Dentin shell fixed (blue arrows) with osteosynthesis screws to the vestibular aspect of the implant. (**b**): The hollow space created between the dentin shell and implant was filled with particulate dentin (green arrows).

**Figure 4 ijerph-18-03174-f004:**
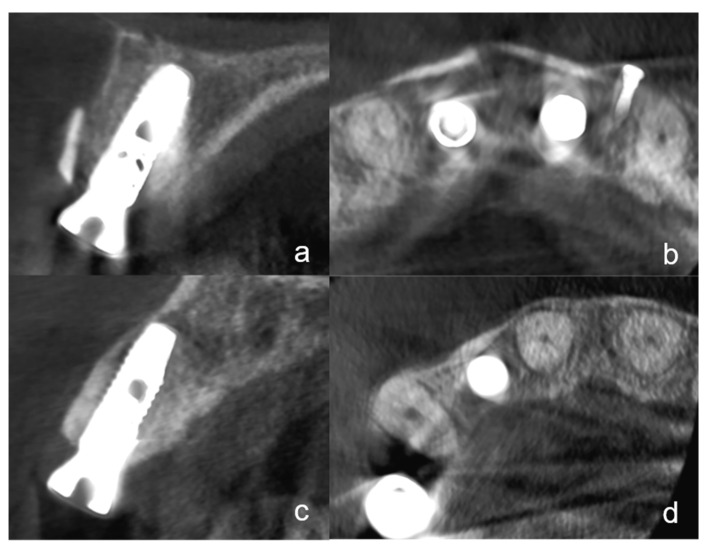
(**a**): A cone-beam computed tomography (CBCT) in the sagittal plane shows an implant regio 11 with the tooth-shell technique (TST) at the time of the implant exposure. The integrity of the buccal lamella is visible. The dentin shell does not appear to show any resorption. (**b**): The same CBCT in the axial plane with the implant in region 11 and another implant region 21. Two buccal dentin shells are clearly visible. (**c**): This figure shows a CBCT in the sagittal plane of an implant region 12 with BST at the time of implant exposure. At this plane, the complete integrity of the buccal lamella can be seen. The bone shell is no longer visible and appears to have undergone replacement resorption. (**d**): The same implant in the CBCT in the axial plane. No bone shell can be seen.

**Table 1 ijerph-18-03174-t001:** Baseline characteristics of the participating patients at the time of augmentation procedure with simultaneous implantation.

		Study Group		Sign.
Baseline Data of Participants	Total	BST	TST	*p*-Value
Age (years)				
Mean (SD)	61.2 (12.7)	60.4 (13.9)	62.0 (11.4)	n.s.
Range	28–82	30–82	28–80	
Gender (male)				
*n* (%)	28 of 59 (47)	15 of 31 (48)	13 of 28 (46)	n.s.

BST = bone-shell technique; TST = tooth-shell technique.

**Table 2 ijerph-18-03174-t002:** Clinical complications at a patient level, region level, and implant level.

		Study Group		Fisher’s Exact Test (2-Sided)
Clinical Complication	Total	BST	TST	*p*-Value
Total severe complications				
*n* (%) on PL	0 of 59 (0)	0 of 31 (0)	0 of 28 (0)	n.s.
*n* (%) on RL	0 of 66 (0)	0 of 32 (0)	0 of 34 (0)	n.s.
*n* (%) on IL	0 of 79 (0)	0 of 41 (0)	0 of 38 (0)	n.s.
Wound dehiscence				
*n* (%) on PL	3 of 59 (5.1)	2 of 31 (6.5)	1 of 28 (3.6)	0.615
*n* (%) on RL	3 of 66 (4.5)	2 of 32 (6.3)	1 of 34 (2.9)	0.519
*n* (%) on IL	3 of 79 (3.8)	2 of 41 (4.9)	1 of 38 (2.6)	0.602
Inflammation (pus)				
*n* (%) on PL	3 of 59 (5.1)	3 of 31 (9.7)	0 of 28 (0)	0.091
*n* (%) on RL	3 of 66 (4.5)	3 of 32 (9.4)	0 of 34 (0)	0.068
*n* (%) on IL	4 of 79 (5.1)	4 of 41 (9.7)	0 of 38 (0)	0.048
Total complications at all				
*n* (%) on PL	6 of 59 (10.2)	5 of 31 (16.1)	1 of 28 (3.6)	0.111
*n* (%) on RL	6 of 66 (9.1)	5 of 32 (15.6)	1 of 34 (2.9)	0.073
*n* (%) on IL	7 of 79 (8.9)	6 of 41 (14.6)	1 of 38 (2.6)	0.061

BST = bone-shell technique; TST = tooth-shell technique; PL = patient level; RL = region level; IL = implant level.

## Data Availability

Data are contained within the article.

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
