# Peer review of "Retrospective Study: Lateral Ridge Augmentation Using Autogenous Dentin: Tooth-Shell Technique vs. Bone-Shell Technique"

_ijerph, 2021, doi:10.3390/ijerph18063174_

Round 1
Reviewer 1 Report
Thank you for giving me this opportunity to review the article entitled, " Proof-of-concept study: lateral ridge augmentation using autogenous dentin: tooth-sell-technique vs. bone-shell-technique".
In general, the article is well structured and has a very interesting methodology.
After reading the manuscript, I have some comments:
- Although the authors say that the ethics committee did not require that patients whose data were collected must be informed (line 413), did they not have to sign a consent form regarding the surgery and its complications?
- Why did the authors use different patients for each technique? Why didn't they use each quadrant of the same patient to evaluate both techniques? If each technique were evaluated on the same patient, but in different quadrants, there would be a decrease in variables that can confuse the results.
- The inclusion criteria are quite incomplete. Lack of description and detail about the state of the oral cavity (periodontitis,...), presence of systemic diseases, medication, if there were teeth adjacent to the implant site,...
- Why is the implants systems used in the inclusion criteria? (line 102) The manuscript is about two lateral ridge augmentation techniques and not about several type of implants.
- The authors do not describe how they arrived at the sample size. Was it referring to the literature consulted or was it a calculation tool?
- Line 139: Complications about who?
- Line 146: " Clinical complications": Are this different definitions scientifically based or were they formulated by the authors?
- Line 283: is missing a B;
- Line 285: Did the inflammation occur in the patient who took more than one implant? May be related to the brushing level? Systemic disease? Medication? If the techniques were evaluated on the same patient, but in different quadrants, could the inflammation not be seen?
- The discussion is well structured, however, they do not explain why TST obtained the observed results. Although they say that dentin has biological properties similar to bone, they don't add much. The discussion should be based on the dentin properties that underlie the observed results.
Author Response
Review 1:
Although the authors say that the ethics committee did not require that patients whose data were collected must be informed (line 413), did they not have to sign a consent form regarding the surgery and its complications?
A: The patients were informed about the surgery, the different possibilities for lateral augmentation (BST and TST) and the individual complications. Patients signed a consent form which is used in routine care. An additional patient consent for retrospective evaluation was not required by the ethics committee.
Why did the authors use different patients for each technique? Why didn't they use each quadrant of the same patient to evaluate both techniques? If each technique were evaluated on the same patient, but in different quadrants, there would be a decrease in variables that can confuse the results.
A: The study was conducted as a retrospective analysis of routine care procedures, where either the TST or the BST was used. This is certainly an interesting approach for later prospective studies. Under these circumstances, such an approach was not possible, also due to the small number of suitable patients with a need for augmentation on both sides.
The inclusion criteria are quite incomplete. Lack of description and detail about the state of the oral cavity (periodontitis,...), presence of systemic diseases, medication, if there were teeth adjacent to the implant site,...
A: As we have already outlined, the present study is a retrospective analysis. Patients with diseases or conditions that contraindicated augmentation (untreated periodontitis, bisphoshonate therapy, radiotherapy,...etc), were not treated even in the course of routine care, so that they were not found in the patient cohort.
Nevertheless, we completed the exclusion criteria according to the review points.
Why is the implants systems used in the inclusion criteria? (line 102) The manuscript is about two lateral ridge augmentation techniques and not about several type of implants.
A: You are completely right, this study is not about different implant types. We have removed these points from the inclusion criteria
The authors do not describe how they arrived at the sample size. Was it referring to the literature consulted or was it a calculation tool?
A: The sample size in this retrospective study is primarily determined by the number of patients treated in the corresponding period of time. However, the sample size exceeds the calculated number of cases of an unfinished prospective study that we are currently conducting and that investigates a similar issue.
Line 139: Complications about who?
A: The list was revised accordingly
Line 146: "Clinical complications": Are this different definitions scientifically based or were they formulated by the authors?
A: The definitions “severe” and “non-severe complications” were formulated by us and are not scientifically based. We have used these terms of complications in other publications before.
- Bartols A, Kasprzyk S, Walther W, Korsch M. Lateral alveolar ridge augmentation with autogenous block grafts fixed at a distance versus resorbable Poly-D-L-Lactide foil fixed at a distance: A single-blind, randomized, controlled trial. Clin Oral Implants Res. 2018;29(8):843-54.
- Korsch M. Tooth shell technique: A proof of concept with the use of autogenous dentin block grafts. Aust Dent J. 2020.
Line 283: is missing a B:
A: Completed
Line 285: Did the inflammation occur in the patient who took more than one implant? May be related to the brushing level? Systemic disease? Medication? If the techniques were evaluated on the same patient, but in different quadrants, could the inflammation not be seen?
A: In total there were 3 Patients (n=3 in BST/ n=0 in TST) with inflammations. One patient with two neighboring implants experienced an inflammation on both implants (one region). The other two patients with inflammations only had a single implant, so that the question you asked cannot be answered. The complications occurred in generally healthy patients without medication. A mechanical irritation(what you described as brushing level) for this could not be identified.
The discussion is well structured, however, they do not explain why TST obtained the observed results. Although they say that dentin has biological properties similar to bone, they don't add much. The discussion should be based on the dentin properties that underlie the observed results.
A: the discussion was revised accordinglyx
Review 1:
Although the authors say that the ethics committee did not require that patients whose data were collected must be informed (line 413), did they not have to sign a consent form regarding the surgery and its complications?
A: The patients were informed about the surgery, the different possibilities for lateral augmentation (BST and TST) and the individual complications. Patients signed a consent form which is used in routine care. An additional patient consent for retrospective evaluation was not required by the ethics committee.
Why did the authors use different patients for each technique? Why didn't they use each quadrant of the same patient to evaluate both techniques? If each technique were evaluated on the same patient, but in different quadrants, there would be a decrease in variables that can confuse the results.
A: The study was conducted as a retrospective analysis of routine care procedures, where either the TST or the BST was used. This is certainly an interesting approach for later prospective studies. Under these circumstances, such an approach was not possible, also due to the small number of suitable patients with a need for augmentation on both sides.
The inclusion criteria are quite incomplete. Lack of description and detail about the state of the oral cavity (periodontitis,...), presence of systemic diseases, medication, if there were teeth adjacent to the implant site,...
A: As we have already outlined, the present study is a retrospective analysis. Patients with diseases or conditions that contraindicated augmentation (untreated periodontitis, bisphoshonate therapy, radiotherapy,...etc), were not treated even in the course of routine care, so that they were not found in the patient cohort.
Nevertheless, we completed the exclusion criteria according to the review points.
Why is the implants systems used in the inclusion criteria? (line 102) The manuscript is about two lateral ridge augmentation techniques and not about several type of implants.
A: You are completely right, this study is not about different implant types. We have removed these points from the inclusion criteria
The authors do not describe how they arrived at the sample size. Was it referring to the literature consulted or was it a calculation tool?
A: The sample size in this retrospective study is primarily determined by the number of patients treated in the corresponding period of time. However, the sample size exceeds the calculated number of cases of an unfinished prospective study that we are currently conducting and that investigates a similar issue.
Line 139: Complications about who?
A: The list was revised accordingly
Line 146: "Clinical complications": Are this different definitions scientifically based or were they formulated by the authors?
A: The definitions “severe” and “non-severe complications” were formulated by us and are not scientifically based. We have used these terms of complications in other publications before.
- Bartols A, Kasprzyk S, Walther W, Korsch M. Lateral alveolar ridge augmentation with autogenous block grafts fixed at a distance versus resorbable Poly-D-L-Lactide foil fixed at a distance: A single-blind, randomized, controlled trial. Clin Oral Implants Res. 2018;29(8):843-54.
- Korsch M. Tooth shell technique: A proof of concept with the use of autogenous dentin block grafts. Aust Dent J. 2020.
Line 283: is missing a B:
A: Completed
Line 285: Did the inflammation occur in the patient who took more than one implant? May be related to the brushing level? Systemic disease? Medication? If the techniques were evaluated on the same patient, but in different quadrants, could the inflammation not be seen?
A: In total there were 3 Patients (n=3 in BST/ n=0 in TST) with inflammations. One patient with two neighboring implants experienced an inflammation on both implants (one region). The other two patients with inflammations only had a single implant, so that the question you asked cannot be answered. The complications occurred in generally healthy patients without medication. A mechanical irritation(what you described as brushing level) for this could not be identified.
The discussion is well structured, however, they do not explain why TST obtained the observed results. Although they say that dentin has biological properties similar to bone, they don't add much. The discussion should be based on the dentin properties that underlie the observed results.
A: the discussion was revised accordingly
Reviewer 2 Report
The authors presented an interesting study on lateral ridge augmentation using autogenous dentin comparing tooth-shell-technique vs. bone-shell-technique. Minor comment on reference check.
Author Response
XCVV
The authors presented an interesting study on lateral ridge augmentation using autogenous dentin comparing tooth-shell-technique vs. bone-shell-technique. Minor comment on reference check.
A: We had no further information on what exactly should be improved. That is why we have concentrated on reviewers 1 and 3 and hope that we have revised everything sufficiently.
Reviewer 3 Report
In this study authors have accomplished a comparative study between tooth-shell-technique (TST) and Bone-shell-technique (BST) using autogenous dentine as an alternative biomaterial to avoid bone harvesting procedures and thus reduce postoperative discomfort of patients. Moreover, within the constrain, authors have found that the TST technique exhibited a safe grafting procedure for lateral alveolar ridge augmentation with conventional result. The authors have reported their findings in the manuscript systematically and results and discussion portion comprised with proper explanation of the finding. I would recommend for publication with minor correction.
- Authors should improve the introduction part with introduction section by adding some advanced dentin materials and other and provide some perspective compared to the autogenous dentin.
- Authors can read and cite following paper https://doi.org/10.3390/ma13143090, https://doi.org/10.3390/nano10091750, https://doi.org/10.3390/nano10071373, and https://doi.org/10.1111/adj.12814
- Figure 1, 2 and 3 are optical image, there is no concrete evidence of the inflammatory effects and complications. How can authors conclude the results based on optical image only?
- I do agree that sometimes clinicians judge the implant situation based on patients feed-back and their physical feelings. But again, authors should analyze and produce qualitative data for proof.
- Authors have discussed the inflammatory effect and another complications, however there is no quantitative data presented?
Author Response
Review 3:
Authors should improve the introduction part with introduction section by adding some advanced dentin materials and other and provide some perspective compared to the autogenous dentin.
A: The introduction part was supplemented by a section about advanced dentin material e.g. ADDM. (Line 74-81)
Authors can read and cite following paper https://doi.org/10.3390/ma13143090, https://doi.org/10.3390/nano10091750, https://doi.org/10.3390/nano10071373, and https://doi.org/10.1111/adj.12814
Figure 1, 2 and 3 are optical image, there is no concrete evidence of the inflammatory effects and complications. How can authors conclude the results based on optical image only?
A: The figures 1 and 3 show the clinical procedure of the BST resp. the TST. The figure 2 shows the processing of the Dentin to obtain the graft. The figures are not intended to show inflammation. Inflammation was defined as the presence of rubor, swelling and/or suppuration and was determined by visual inspection. In addition, at time of implant exposure a peri-implant tissue probing at four aspects and radiographic evaluation of the peri-implant hard tissues were performed.
I do agree that sometimes clinicians judge the implant situation based on patients feed-back and their physical feelings. But again, authors should analyze and produce qualitative data for proof.
A: Patient reported outcome measures were not surveyed in this study.
Authors have discussed the inflammatory effect and another complications, however there is no quantitative data presented?
A: The incidence of complications is displayed in Table 2